# High-Throughput Mechanistic Screening of Epigenetic Compounds for the Potential Treatment of Meningiomas

**DOI:** 10.3390/jcm10143150

**Published:** 2021-07-16

**Authors:** Philip D. Tatman, Tadeusz H. Wroblewski, Anthony R. Fringuello, Samuel R. Scherer, William B. Foreman, Denise M. Damek, Kevin Lillehei, A. Samy Youssef, Randy L. Jensen, Michael W. Graner, D. Ryan Ormond

**Affiliations:** 1Department of Neurosurgery, Anschutz Medical Campus, University of Colorado, Aurora, CO 80045, USA; philip.tatman@cuanschutz.edu (P.D.T.); tadeusz.wroblewski@cuanschutz.edu (T.H.W.); anthonyfringuello@gmail.com (A.R.F.); samuelrscherer@gmail.com (S.R.S.); will.b.foreman@outlook.com (W.B.F.); DENISE.DAMEK@CUANSCHUTZ.EDU (D.M.D.); KEVIN.LILLEHEI@CUANSCHUTZ.EDU (K.L.); SAMY.YOUSSEF@CUANSCHUTZ.EDU (A.S.Y.); 2Medical Scientist Training Program, Anschutz Medical Campus, University of Colorado, Aurora, CO 80045, USA; 3Department of Pharmacology, Anschutz Medical Campus, University of Colorado, Aurora, CO 80045, USA; 4Department of Neurology, Anschutz Medical Campus, University of Colorado, Aurora, CO 80045, USA; 5Department of Neurosurgery, Huntsman Cancer Institute, University of Utah, Salt Lake City, UT 84112, USA; Randy.Jensen@hsc.utah.edu

**Keywords:** meningioma, HDAC inhibitors, panobinostat, LAQ824, HC toxin, high-throughput screening

## Abstract

Background: Meningiomas are the most common primary central nervous system tumors. 20–30% of these tumors are considered high-grade and associated with poor prognosis and high recurrence rates. Despite the high occurrence of meningiomas, there are no FDA-approved compounds for the treatment of these tumors. Methods: In this study, we screened patient-cultured meningiomas with an epigenetic compound library to identify targetable mechanisms for the potential treatment of these tumors. Meningioma cell cultures were generated directly from surgically resected patient tumors and were cultured on a neural matrix. Cells were treated with a library of compounds meant to target epigenetic functions. Results: Although each tumor displayed a unique compound sensitivity profile, Panobinostat, LAQ824, and HC toxin were broadly effective across most tumors. These three compounds are broad-spectrum Histone Deacetylase (HDAC) inhibitors which target class I, IIa, and IIb HDACs. Panobinostat was identified as the most broadly effective compound, capable of significantly decreasing the average cell viability of the sample cohort, regardless of tumor grade, recurrence, radiation, and patient gender. Conclusions: These findings strongly suggest an important role of HDACs in meningioma biology and as a targetable mechanism. Additional validation studies are necessary to confirm these promising findings, as well to identify an ideal HDAC inhibitor candidate to develop for clinical use.

## 1. Introduction

Meningiomas comprise 33.8% of all primary brain tumors, making them the most common primary central nervous system (CNS) tumors [1]. From 20–30% of meningiomas are classified by the World Health Organization (WHO) as grade II or III meningiomas, which have five-year recurrence rates of 38–55% and 79%, respectively [2,3,4,5]. While higher-grade meningiomas have poor outcomes overall, it has been previously reported that patients with grade I meningiomas also experience long-term neurological deficits and reduced overall survival [6]. These tumors lack a United States Food and Drug Administration (FDA)-approved pharmacotherapeutic agent [7]. Surgery remains the primary treatment modality for accessible, symptomatic meningiomas, often followed by radiation therapy. Treatment options for recurrent meningiomas after maximally feasible surgery and radiation are limited, with minimal efficacy shown to date from chemotherapy [7,8]. The paucity of treatment options and poor outcomes for patients with meningiomas necessitate additional research into targetable mechanisms for the development of novel therapies.

Previous work on meningioma genetics has yielded little clinical benefit, despite significant findings. *NF2* (neurofibromatosis 2) is the most common mutation in meningiomas, occurring in 40–60% of sporadic meningiomas, and has been implicated as a driver of meningioma tumorigenesis [9,10]. Two independent studies have identified *TRAF7* (TNF receptor-associated factor 7), *AKT1* (v-akt murine thymoma viral oncogene homolog 1), and *SMO* (Smoothened, frizzled family receptor) mutations as drivers of meningioma oncogenesis [11,12]. Mutations in *TRAF7* have been identified in nearly one-fourth of meningiomas, while mutations in *AKT1* and *SMO* have lower prevalence and are associated with higher grade *NF2* wild-type tumors [12]. *POLR2A* (RNA Polymerase II Subunit A) mutations in meningiomas have also been identified, being linked to a meningothelial histology and tuberculum sellae tumors [13]. More recently, mutations in *TERT* (telomerase reverse transcriptase) and *CDKN2A* (cyclin-dependent kinase inhibitor 2A) have been shown to be potential prognostic indicators in higher grade meningiomas [14,15]. Mutations in the *TERT* promoter have been associated with worse prognosis and decreased survival rates in grade II and III meningiomas, and *CDKN2A* mutation or gene deletion has been associated with meningioma recurrence [15,16,17,18]. Although the mutational landscape of meningiomas is becoming more well defined, these findings have not yet led to additional, or improved, therapies.

A plausible solution to the lack of pharmacotherapies for the treatment of meningiomas is to directly measure their drug sensitivities. Screening meningiomas against a compound library enables direct insight into targetable mechanisms as well as potential candidate compounds, depending on library construction. With respect to compound library identification, a promising class of mechanisms with the potential to treat meningiomas is epigenetics. Epigenetics are defined as all processes that regulate structure and access to DNA resulting in transcriptional changes, DNA replication, and DNA repair [19,20,21]. These are described as “writers” (which add chromatin and histone modifications), “erasers” (which remove modifications), and “readers” (which recognize modifications and mediate effects) of epigenetics [22]. Previous studies have identified epigenetic dysregulation as a driver of meningioma oncogenesis and recurrence [23,24,25,26]. CpG island and promoter DNA methylation profiles are known to vary with meningioma pathophysiology and clinical outcomes [24,25,26,27]. Furthermore, meningioma whole-genome DNA methylation profiles correlate more accurately to clinical prognosis, compared with the current WHO tumor grading system. Additionally, histone deacetylases (HDACs) have been implicated in meningiomas through the regulation of transcriptional changes, downstream of *PI3K/Akt*, in the setting of *NF2* deletion [9,28,29,30]. The connection between epigenetic dysregulation of meningiomas and clinical outcomes warrants a comprehensive investigation into the targeting of epigenetic mechanisms as a viable treatment avenue for meningiomas.

To identify targetable epigenetic mechanisms for the treatment of meningiomas, we performed a high throughput drug screen of 32 patient-cultured meningiomas against a 139-compound epigenetic library. The compounds in the library were designed to modulate activities of various classes of epigenetic effectors as a mechanistic-based screen. The cultured meningiomas were screened within 3 weeks of surgical resection, establishing a rapid means to identify potential drug sensitivities for therapeutic exploitation. Among the different classes of targets, multiple drugs that target HDACs were identified as having the ability to significantly inhibit meningioma growth, suggesting that HDAC inhibition is a promising therapeutic modality for the treatment of meningiomas.

## 2. Materials and Methods

### 2.1. Patient Identification, Demographics, and Tumor Collection

Patients undergoing surgery for meningiomas were identified through the Neurosurgery Nervous System Biorepository at the University of Colorado, Anschutz Medical Campus. From a period of September 2018 to May of 2019, tumor specimens were collected at the time of surgery and transported directly to our lab for tissue culture. In accordance with IRB protocol (IRB #13-3007), patients were consented to that protocol, and were deidentified. Established patient-derived meningioma cell lines, provided by Dr. Randy Jensen from the University of Utah (IRB #00010924), were also included in this study. For each tumor, we recorded relevant demographic information including patient age and gender, as well as tumor characteristics including grade, histologic subtype, primary vs. recurrent, and history of radiation and chemotherapy.

### 2.2. Tissue Culture

Meningiomas were collected from the operating rooms at the University of Colorado Anschutz Medical Campus and sectioned into 1 mm × 1 mm pieces in sterile PBS (pH 7.4), under sterile tissue culture hood conditions. The tumors were then digested with dispase into single-cell suspensions and plated in flasks containing a thin film of decellularized bovine neural matrix [31]. Meningiomas were cultured in high glucose DMEM with 15% FBS, 1% pen/strep, and 1% Glutamax and passaged weekly until used. All tumors were screened within three weeks of surgical resection and never cryogenically frozen, with the exception of tumors originating from the University of Utah.

### 2.3. Epigenetic Compound Panel Screening

Epigenetic compound screening was completed with a 139-compound epigenetic library (*Cayman Chemical, Ann Arbor, MI, USA*). Cells were plated into neural-matrix coated 96-well plates and seeded at a density of 2500 cells per well. After 24 h in culture, individual epigenetic compounds were added to individual wells to a final concentration of 1 µM. Seventy-two hours after the addition of the library, cell viability was determined via MTS assay. Each compound was tested in triplicate for each tumor.

### 2.4. Cell Viability

The cell viability assay was performed with a MTS tetrazolium assay, as previously described [32]. The MTS reagent was prepared with PBS (pH 7.4), to which MTS powder (2 mg/mL) and PES powder (0.21 mg/mL) were added. The MTS reagent was added to each well for a final MTS concentration of 0.33 mg/mL. Absorbance was recorded at 490 nm in a Biotek plate reader (BioTek, Winooski, VT, USA).

### 2.5. Statistics

All statistics were calculated in the *R statistical suite (version 4.0.2) (https://www.r-project.org/)*. All plates were background subtracted and normalized to vehicle controls, where the vehicle control was comprised of 0.01% DMSO in media. Significantly effective compounds for individual tumors were identified by filtering for *p* values less than 0.05, calculated using the Mann–Whitney U test (confidence level of 0.95). Broadly effective compounds across the entire cohort, or given subgroup of tumors, were identified using the Student’s *t* test (confidence level of 0.95) with significance determined by a *p* value less than 0.05. To determine the significance between three or more subgroups of tumors, an ANOVA with a Tukey HSD post-hoc test (confidence level of 0.95) was conducted with a *p* value of less than 0.05. Data are displayed as the mean +/− standard deviation.

## 3. Results

### 3.1. Patient Demographics and Tumor Information

Thirty-two tumors were cultured from patients, four of which were from Utah and are indicated as such with an asterisk in Table 1. The mean age of patients in our cohort at the time of surgery was 57 +/− 12.5 SD years, ranging from 38–95 years (Table 1). A total of 21.9% of the patients were male (*n = 7*) and 78.1% female (*n = 25*, Table 1). The cohort of tumors consisted of 25 grade I tumors, 6 grade II tumors, and 1 grade III tumor, including 3 recurrent and 29 primary tumors (Table 1). The tumors were further separated by histologic subtype, history of radiation, and chemotherapy as described in Table 1**.**

### 3.2. Meningiomas Are Broadly Sensitive to Epigenetic Inhibition and Have Unique Drug Sensitivity Profiles

Screening our cohort of patient-cultured meningiomas demonstrated that epigenetic compounds can significantly decrease cell viability (Figure 1a). On average, 27 +/− 15.3 SD compounds significantly reduced cell viability per tumor (Appendix A). Across all tumors, cell viability was reduced from 100% to 75% in 29/32 tumors, to 50% in 24/32 tumors, and to 25% in 3/32 tumors (Figure 1a, Appendix A), clearly showing that meningiomas can be inhibited by epigenetic compounds.

Broadly effective compounds were identified as those that reduced cell viability to an average of 80%, or less, across our tumor cohort (not necessarily individual tumors), as well as a *p* value of less than 0.05 (Figure 1b, Appendix A). This screen identified panobinostat (51.9% +/− 23.9 SD, *p* = 1.06 × 10^−40^); LAQ824 (56.1% +/− 24 SD, *p* = 2.16 × 10^−36^); HC Toxin (59.4% +/− 24.7 SD, *p* = 3.16 × 10^−32^); gemcitabine (71.4% +/− 22.3 SD, *p* = 6.58 × 10^−25^); JIB-04 (77.3% +/− 28.9 SD, *p* = 4.60 × 10^−13^); and SB939 (79.6% +/− 23.7 SD, *p* = 9.30 × 10^−15^). HC Toxin significantly reduced cell viability in 27/32 tumors, panobinostat reduced cell viability in 26/32 tumors, and LAQ824 and gemcitabine reduced cell viability in 25/32 tumors (Figure 1b). Only one tumor out of this cohort did not have a significant reduction in cell viability after epigenetic inhibition (Appendix A). These data demonstrate the common biology among meningiomas based on their sensitivity to a small cohort of drugs, as well as the potential to target a common mechanism in meningiomas.

To identify compounds specific to each tumor, we rank ordered compounds that significantly reduced cell viability by a *p* value less than 0.05 and displayed the top five for each tumor (Figure 2, Appendix A). This analysis highlights the most effective compounds per individual tumor. Interestingly, panobinostat was the most effective single agent in 44% (14/32) of tumors, followed by JIB-04 in 19% (6/32), LAQ824 in 13% (4/32), HC Toxin in 6% (2/32), OTX015 in 6% (2/32), and UNC0631 in 6% (2/32). Despite meningiomas having unique drug sensitivity profiles (Figure 1a), they share a high degree of sensitivity to a small group of compounds (Figure 2). Unsurprisingly, of the broadly effective compounds reported in the previous paragraph, panobinostat appeared in 75% (24/32) of the top five most effective compounds per tumor, LAQ824 in 72% (23/32), HC Toxin in 66% (21/32), gemcitabine in 34% (11/32), JIB-04 in 34% (11/32), and SB939 in 9% (3/32). While the unique sensitivity profiles of individual meningiomas suggest that a personalized approach to treatment may be most efficacious (Figure 1a, Figure 2), this analysis shows that most meningiomas in our screen are sensitive to a small number of compounds.

### 3.3. Influence of Grade on the Use of Epigenetic Compounds for the Treatment of Meningiomas

We next examined the differences in compound sensitivity with respect to tumor grade (Appendix A). The most broadly effective compound for grade I tumors was panobinostat, with an average cell viability of 51.9% +/− 24.8 SD (*p* = 1.05 × 10^−30^, Figure 3a, Appendix A), and, for grade II tumors, the most broadly effective compound was LAQ824, with an average cell viability of 44.8% +/− 16.8 SD (*p* = 1.67 × 10^−12^), followed by panobinostat with 52.7% +/− 19.1 SD (*p* = 3.04 × 10^−10^, Figure 3b, Appendix A). LAQ824 was significantly more effective in grade II compared with grade I tumors (*p* = 0.0228), and there were no significant differences between the effectiveness of panobinostat comparing grade I and grade II tumors (Appendix A). Panobinostat, LAQ824, HC Toxin, gemcitabine, and JIB-04 were all significantly effective at reducing the average cell viability in both grade I and grade II tumors (Figure 3a,b, Appendix A). Our single grade III tumor demonstrated unique sensitivities with the most effective compounds including UNC0631 (22.1% +/−1.2 SD, *p* = 1.82 × 10^−35^); UNC0646 (22.4% +/−2.9 SD, *p* = 1.39 × 10^−8^); MI-NC hydrochloride (24.2% +/− 7.4 SD, *p* = 1.33 × 10^−4^); HC Toxin (27.6% +/− 1.0 SD, *p* = 1.09 × 10^−27^); and SB939 (28.9% +/− 4.2 SD, *p* = 3.50 × 10^−6^, Figure 3c, Appendix A). Notably, the grade III tumor displayed significant sensitivity to many of the compounds found to be efficacious across all meningiomas, including panobinostat resulting in 46.6% +/− 33.5 SD average cell viability (*p* = 0.0496) and LAQ824 with 39.1% +/− 31.7 SD average cell viability (*p* = 0.0125). These findings implicate that each grade of meningioma has a different sensitivity profile, though all three grades remain sensitive to a small cohort of broadly effective compounds.

To identify the individual sensitivities of each tumor in the respective grades, we repeated the top most effective compound analysis (Figure 3d). Panobinostat was the most effective compound in 52% (13/25) of grade I tumors, followed by JIB-04 in 16% (4/25). For grade II tumors, JIB-04 and LAQ824 were each the most effective compound in 33% of tumors, inhibiting 2/6 different tumors. UNC0631 was the most effective compound for the grade III tumor.

### 3.4. Efficacy of Epigenetic Compounds on Primary vs. Recurrent Meningiomas

We next sought to understand the compound sensitivity differences between primary and recurrent tumors and found that the effectiveness of individual compounds is largely preserved across both etiologies (Figure 4a,b, Appendix A). Panobinostat was the most broadly effective compound for both primary and recurrent meningiomas resulting in an average cell viability of 52.1% +/−24.3 SD (*p* = 2.34 × 10^−36^) for primary tumors and 49.0% +/− 20.4 SD in recurrent tumors (*p* = 2.28 × 10^−5^). Primary tumors also exhibited sensitivities to LAQ824 (56.2% +/− 23.2 SD, *p* = 6.62 × 10^−34^), HC Toxin (59.2% +/− 24.3 SD, *p* = 2.88 × 10^−30^), gemcitabine (71.5% +/− 21.0 SD, *p* = 2.11 × 10^−24^), and JIB-04 (78.2% +/− 28.1 SD, *p* = 1.09 × 10^−11^); recurrent tumors were broadly sensitive to LAQ824 (55.0% +/− 31.5 SD, *p* = 7.91 × 10^−4^), HC Toxin (60.9% +/− 30.2 SD, *p* = 4.64 × 10^−3^), apicidin (61.9% +/− 28.9 SD, *p* = 2.41 × 10^−3^), and SB939 (62.7% +/− 29.9 SD, *p* = 3.37 × 10^−3^). Notably, no significant difference in effectiveness between these compounds was found between primary and recurrent tumors (Appendix A), suggesting the potential to treat these two etiologies by targeting similar mechanisms.

The five most effective compounds for either primary or recurrent tumors were also identified (Figure 4c). Panobinostat was the number one most effective compound for 45% (13/29) of primary tumors and 33% (1/3) of recurrent tumors; JIB-04 was the most effective compound for 17% (5/29) primary tumors and 33% (1/3) of recurrent tumors; and LAQ824 was the most effective compound in 14% (4/29) of primary tumors. These data suggest that regardless of tumor recurrence, a small cohort of compounds are effective in most meningiomas.

### 3.5. Meningioma Sensitivity to Epigenetic Inhibition Based on Radiation History

To identify the effect of prior radiation on meningioma sensitivity to epigenetic inhibition, we grouped our cohort by tumors with a prior history of radiation, no radiation, or not specified (Appendix A). Broadly effective compounds were determined, as described above, and identified for each of the radiation history categories (Figure 5a–c, Appendix A). Tumors with no history of radiation were broadly sensitive to panobinostat (52.5% +/− 23.1 SD, *p* = 7.82 × 10^−35^), LAQ824 (59.0% +/− 23.0 SD, *p* = 6.51 × 10^−30^), HC Toxin (62.7% +/− 24.6 SD, *p* = 9.51 × 10^−26^), gemcitabine (73.8% +/−18.7 SD, *p* = 1.38 × 10^−23^), and JIB-04 (77.7% +/− 28.6 SD, *p* = 4.57 × 10^−11^). The single tumor with prior radiation, which notably was not radiation induced, was significantly inhibited by JIB-04 (25.7% +/− 9.6 SD, *p* = 4.3 × 10^−3^), panobinostat (31.8% +/− 1.4 SD, *p* = 2.81 × 10^−16^), LAQ824 (35.8% +/− 8.5 SD, *p* = 4.18 × 10^−3^), HC toxin (42.2% +/− 8.4 SD, *p* = 5.22 × 10^−3^), and CPI-203 (43.5% +/− 1.3 SD, *p* = 7.50 × 10^−18^). The radiation non-specified tumors were sensitive to HC toxin (35.7% +/− 9.2 SD, *p* = 3.07 × 10^−13^), SB939 (39.0% +/− 11.3 SD, *p* = 6.2 × 10^−14^), UNC0646 (40.5% +/− 27.1 SD, *p* = 1.22 × 10^−6^), and LAQ824 (41.5% +/− 26 SD, *p* = 8.74 × 10^−7^); those tumors were also significantly inhibited by previously mentioned broadly effective compounds, panobinostat (52.2% +/− 30.1 SD, *p* = 2.20 × 10^−5^) and gemcitabine (59.5% +/−37.7 SD, *p* = 1.47 × 10^−3^).

When analyzing the most effective individual compounds per tumor, grouped by radiation history (Figure 5d, Appendix A), we found in tumors with no prior history of radiation that panobinostat was the most effective compound in 51.9% (14/27) of tumors, followed by JIB-04 in 18.5% (5/27) of tumors, and LAQ824 in 11.1% (3/27) of tumors. In our single tumor with prior radiation, the most effective compound was JIB-04. Finally, in our cohort of four tumors with non-specified radiation history, each tumor responded best to a different compound, which included LAQ824, HC Toxin, UNC0631, and CAY-10398.

### 3.6. Impact of Gender on Epigenetic Inhibition of Meningiomas

We next compared the sensitivity of meningiomas to epigenetic inhibition between tumors from female and male patients (Appendix A). The most effective compound for female-derived patient tumors was panobinostat with an average cell viability of 53.1% +/− 23.7 SD (*p* = 4.10 × 10^−31^), followed by LAQ824 (58.4% +/− 23.9 SD, *p* = 3.75 × 10^−27^), and HC toxin (62.8% +/− 24.8 SD, *p* = 1.10 × 10^−23^, Figure 6a, Appendix A). HC toxin was the most broadly effective compound in male-derived tumors, demonstrating an average cell viability of 45.3% +/− 19.3 SD (*p* = 1.14 × 10^−11^), followed by panobinostat (47.6%, +/− 24.8 SD, *p* = 9.62 × 10^−11^) and LAQ824 (48.1% +/− 23.2 SD, *p* = 7.08 × 10^−11^, Figure 6b, Appendix A). Comparing the top five compounds from the tumors originating from male and female patients, there was no significant difference in the efficacy of panobinostat, LAQ824, and gemcitabine (Appendix A). JIB-04 was significantly more effective in female patients than male patients (*p* = 0.0279), while HC Toxin (*p* = 1.19 × 10^−3^), SB939 (*p* = 3.20 × 10^−6^), and 6-thioguanine (*p* = 9.56 × 10^−6^) were significantly more effective in male patients than female patients (Appendix A).

We then assessed the most effective compound per tumor, separated by gender (Figure 6c). Panobinostat was the most effective compound in 44% (11/25) of tumors from female patients, followed by JIB-04 in 24% (6/25) of tumors, and LAQ824 in 12% (3/25) of tumors. Panobinostat was also the most effective compound in 43% (3/7) of male-derived patient tumors, while the other four male tumors were uniquely most sensitive to individual compounds, including BIX01294 (hydrochloride hydrate), HC Toxin, LAQ824, and UNC0631.

## 4. Discussion

In this study, we present a 32-meningioma cohort screened with 139 epigenetic compounds to find targetable mechanisms for the treatment of meningiomas. While all meningiomas displayed a unique sensitivity profile, most were sensitive to a small group of compounds. The most broadly effective compounds identified—panobinostat, LAQ824 and HC toxin—all act via HDAC inhibition. Panobinostat and LAQ824 target class I, class IIa and class IIb HDACs [33,34], while HC-toxin is a reversible, cell permeable HDAC inhibitor [35,36]. The two exceptions to this were the single tumor with known prior radiation (Figure 5b), which exhibited greater sensitivity to a Jumanji domain inhibitor, and the single grade 3 tumor (Figure 3c), which showed greater sensitivity to G9a/GLP inhibition, although both of these tumors remained sensitive to HDAC inhibition. These data suggest a common dependence on HDAC function for the viability of meningiomas independent of grade, prior radiation, recurrence, or patient gender; therefore, HDACs present attractive therapeutic targets.

HDACs are an important epigenetic mechanism in meningiomas, in part, because of their regulation of the *Akt* pathway, which is affected by the inactivation of *NF2* [9,11,37]. Previous studies, utilizing meningioma mouse xenograft models, have implicated the efficacy of pan-HDAC inhibitors [38,39]. In a 2013 study, the investigators evaluated pan-HDAC inhibition with compound AR-42 in Ben-Men cell-line intracranial xenografts and found a significant reduction in tumor growth [38]. In a more recent study, investigators using a patient-derived orthotopic xenograft model achieved a significant reduction in tumor growth using panobinostat [39].

One of the primary limitations of this study was the use of a single dose of drug across a large library of compounds. The reason we limited this study to a single dose was due to the low number of tumor cells available, which were spread across treatments by 139 compounds. In doing so, we were able to ensure our ability to screen the entire library in triplicate for every tumor. The 1 mM dose was selected based on a study by Liston and Davis, which identified 1.47 mM as the median maximum plasma concentration in humans across a cohort of 145 FDA-approved cancer compounds [40]. The limited number of tumor cells also prevented further analysis of the nature of cell death/cytotoxicity versus cytostasis, as we relied on MTS assays to assess drug-driven cell growth inhibition. MTS assays are facile, reproducible, and relatively inexpensive for large-scale screens [32], and we chose this as our primary indicator of drug utility because it is compatible with our culture system.

The total cohort exceeded 30 tumors and therefore was an appropriate size to apply standard population statistics; however, a significant limitation is the lack of population to appropriately power sub-analyses. Nonetheless, this cohort reflects the generalized epidemiologic trends of the disease: in terms of ratios of tumor grades, 80–85% grade 1 tumors are reported; our cohort was ~79% grade 1 tumors; 15–20% grade 2 tumors are reported; our cohort was ~16% grade 2; 1–2% grade 3 tumors are reported; our single grade 3 tumor would represent ~3% [41,42]. Recurrence rates vary according to tumor grade and extent of resection, but 5-year recurrence rates range from as low as 7% (grade 1) to as high as 78% (grade 3) [42]. Without subclassification according to patients’ previous surgical and clinical parameters, and reflecting our short time course of tumor collection (~9 months), < 10% of our cohort presented with recurrent tumors, and no individuals with primary tumors within our cohort recurred during the course of this study. Radiation therapy likewise varies as a treatment, mostly for sub-total resections and higher tumor grades, while most grade 1 tumors are not radiated [41,42], making it difficult to generalize. Our cohort had only one patient known to have received radiation and four cases where this was not specified. Female preponderance of the disease is reported to be 2.27:1 [41,42], and our cohort was ~3.5:1, female to male. Thus, ratios of tumor grades, female-to-male skew of patients, rates of recurrence, and use radiation therapy among our cohort are remarkably similar to those categories found in larger population studies. We do realize that to achieve sufficient numbers of particular tumor subsets for a more robust statistical evaluation (e.g., tumors of higher grade, tumors that are recurrent, or those that have received prior radiation), we will need much longer collection periods and collaborations with other institutions.

The findings of this study present HDAC inhibition as a promising treatment avenue for meningioma. However, as this was a mechanistic screen for classes of epigenetic compounds, further validation studies are still required to identify an ideal clinical candidate compound for further validation in pre-clinical studies. Although the lack of mechanistic studies makes it unclear as to how meningiomas are dependent on HDAC activity, the suppressed growth observed in our in vitro cohort as well as these two in vivo studies [38,39] supports the therapeutic potential of HDAC inhibitors for the treatment of meningioma.

## Figures and Tables

**Figure 1 jcm-10-03150-f001:**
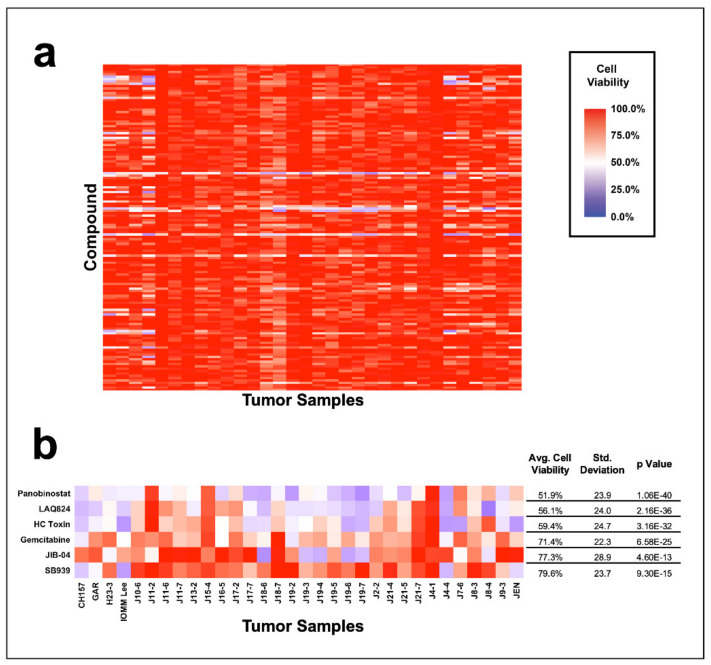
Meningioma sensitivity to the epigenetic compound library. (**a**) Heat map of meningiomas screened against 139-compound epigenetic panel. (**b**) Heat map of broadly effective compounds, identified as compounds that significantly decrease the average cell viability of the cohort to 80% or lower and have a *p* value less than 0.05.

**Figure 2 jcm-10-03150-f002:**
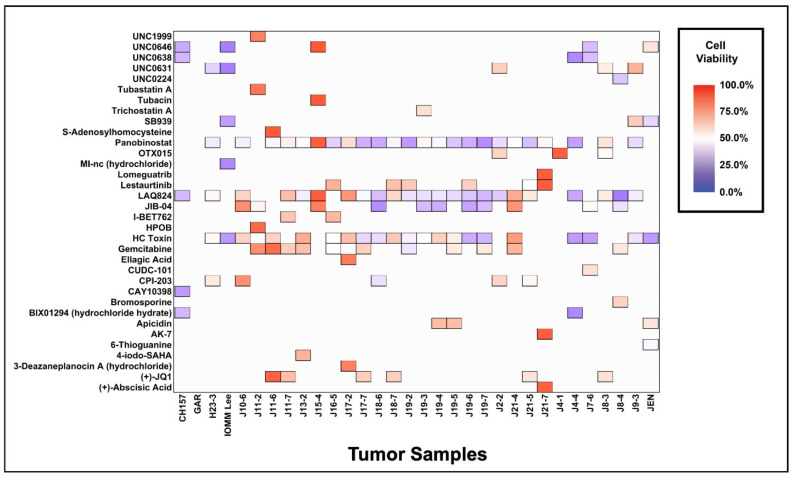
Top five most effective epigenetic compounds per tumor. Heat map of the most effective compounds, determined as the top five compounds that reduce cell viability the most for each tumor. The top five compounds are displayed as filled tiles in the heat map, while compounds that were not in the top five remain as white space.

**Figure 3 jcm-10-03150-f003:**
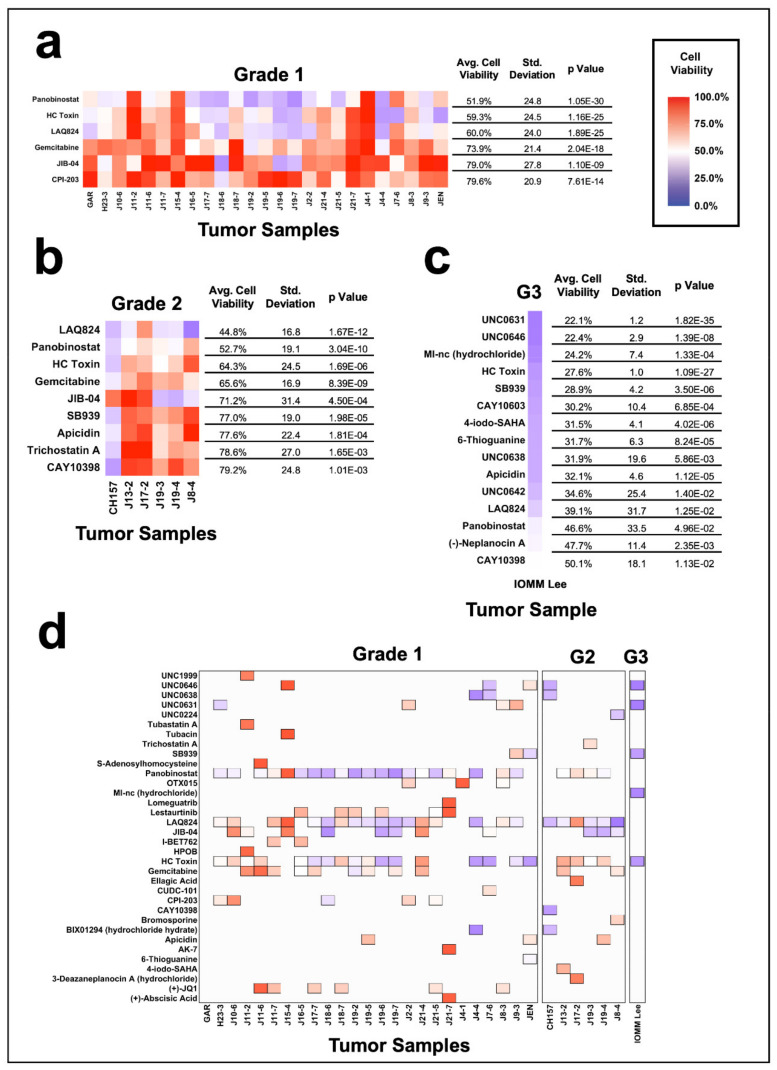
Comparison of the meningioma cohort by tumor grade. Heat maps displaying the broadly most effective compound separated by tumor grade: (**a**) grade 1 tumors, (**b**) grade 2 tumors, and (**c**) grade 3 tumors, denoted as “G3”. Broadly effective compounds for each grade are identified as compounds that reduce the average cell viability of the group to 80% or less and have a *p* value of less than 0.05. Heat maps were limited to the top 15 most effective compounds when applicable. (**d**) Heat map of the top five most effective epigenetic compounds, faceted by grade 1, grade 2 (denoted as “G2”), and grade 3 (denoted as “G3”) meningiomas. The top five compounds are displayed as filled tiles in the heat map, while compounds that were not in the top five remain as white space.

**Figure 4 jcm-10-03150-f004:**
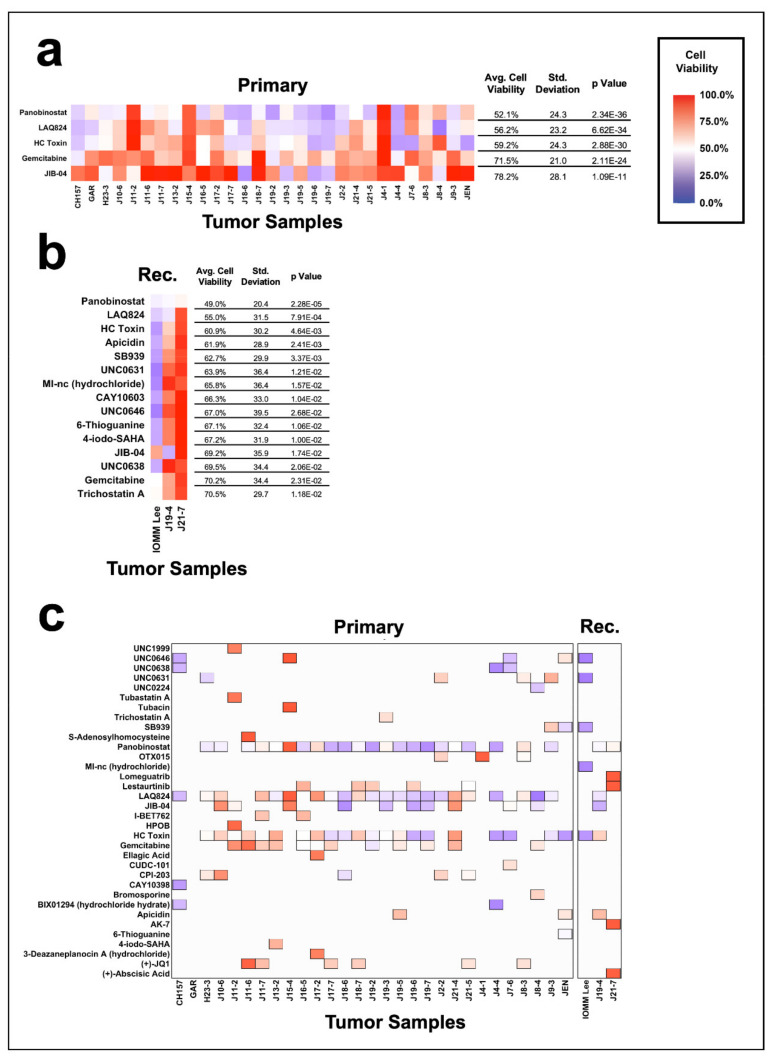
Comparison of the meningioma cohort separated by primary and recurrent tumors. Heat maps displaying the broadly effective compounds for (**a**) primary meningiomas and (**b**) recurrent (denoted as “Rec”) meningiomas. Broadly effective compounds for primary and recurrent meningiomas are identified as compounds that reduce the average cell viability of the group to 80% or less and have a *p* value of less than 0.05. Heat maps were limited to the top 15 most effective compounds when applicable. (**c**) Heat map of the top five most effective epigenetic compounds, separated by primary and recurrent (denoted as “Rec.”) meningiomas. The top five compounds are displayed as filled tiles in the heat map, while compounds that were not in the top five remain as white space.

**Figure 5 jcm-10-03150-f005:**
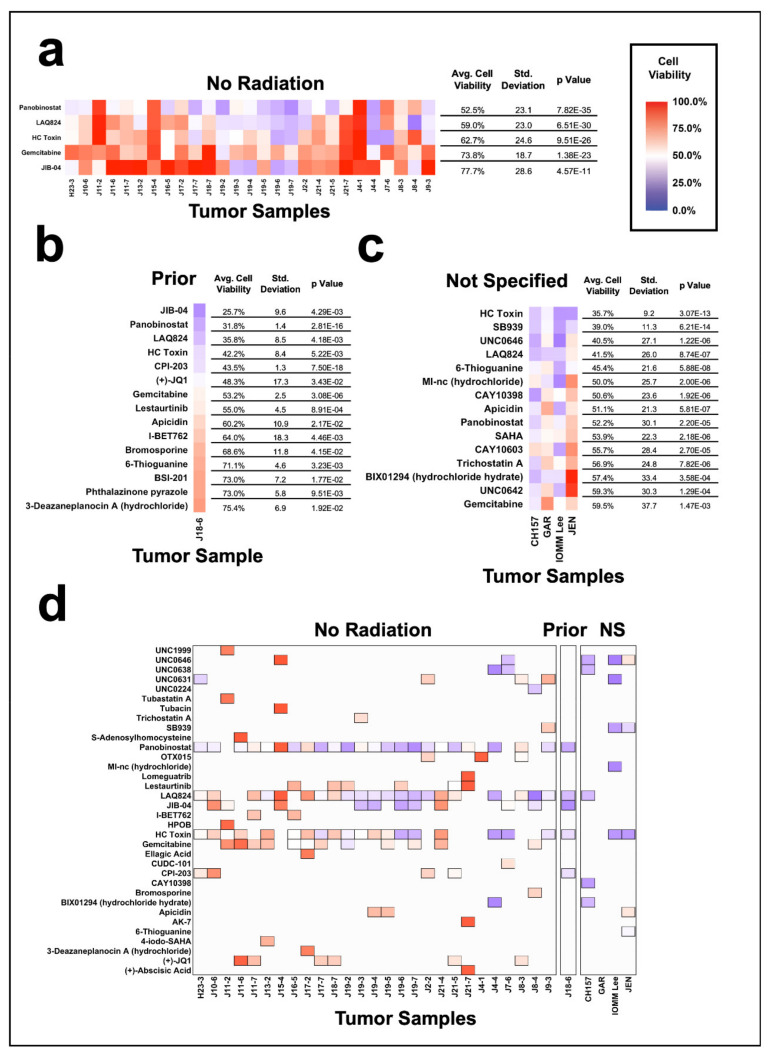
Comparison of the meningioma cohort by history of tumor radiation. Heat maps displaying the broadly most effective compounds for meningiomas with (**a**) no history of radiation, (**b**) prior history of radiation (denoted as “Prior”), and (**c**) tumors with a non-specified history of radiation. Broadly effective compounds for tumors with no radiation, prior radiation, or non-specified radiation are identified as compounds that reduce the average cell viability of the group to 80% or less and have a *p* value of less than 0.05. Heat maps were limited to the top 15 most effective compounds when applicable. (**d**) Heat map of the top five most effective epigenetic compounds, facetted by meningiomas with no radiation, non-specified history of radiation (denoted as “NS”), and tumors with prior radiation (denoted as “Prior”). The top five compounds are displayed as filled tiles in the heat map, while compounds that were not in the top five remain as white space.

**Figure 6 jcm-10-03150-f006:**
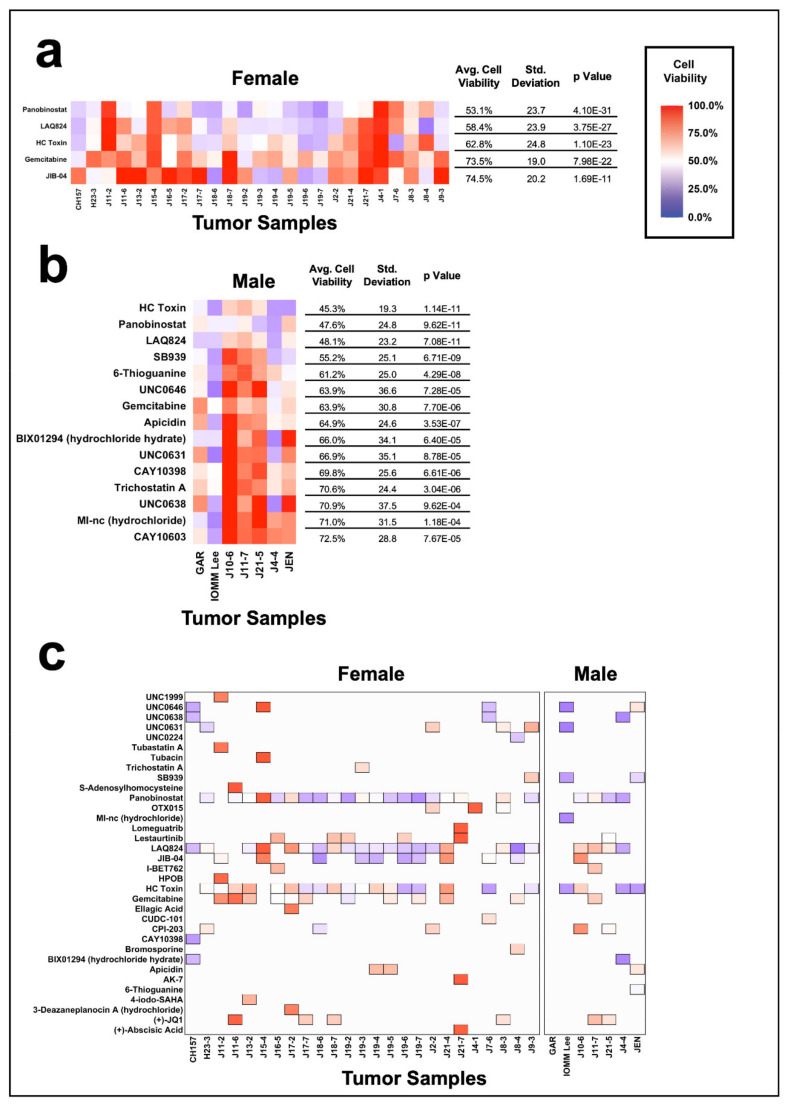
Comparison of the meningioma cohort by patient gender. Heat maps of broadly effective compounds, separated by tumors originating from (**a**) female patients and (**b**) male patients. Broadly effective compounds for meningiomas derived from female and male patients are identified as compounds that reduce the average cell viability of the group to 80% or less and have a *p* value of less than 0.05. (**c**) Heat map of the top five most effective epigenetic compounds, separated by female and male patients. The top five compounds are displayed as filled tiles in the heat map, while compounds that were not in the top five remain as white space.

**Table 1 jcm-10-03150-t001:** Patient demographics for the meningioma cohort.

*No.*	Sample	Age	Gender	Grade	Histologic Subtype	Primary/Recurrent	Radiation	Chemotherapy
*1*	*IOMM Lee	61	Male	3	Anaplastic	Recurrent	Unknown	Unknown
*2*	*CH157	59	Female	2	Unknown	Primary	Unknown	Unknown
*3*	J8-4	48	Female	2	Atypical	Primary	No	No
*4*	J13-2	54	Female	2	Atypical	Primary	No	No
*5*	J17-2	68	Female	2	Atypical	Primary	No	No
*6*	J19-3	95	Female	2	Atypical	Primary	No	No
*7*	J19-4	49	Female	2	Atypical	Recurrent	No	No
*8*	*GAR	71	Male	1	Unknown	Primary	Unknown	No
*9*	H23-3	83	Female	1	Small cell with psammoma bodies	Primary	No	No
*10*	J10-6	54	Male	1	Meningothelial	Primary	No	No
*11*	J11-2	57	Female	1	Not Specified	Primary	No	No
*12*	J11-6	56	Female	1	Not Specified	Primary	No	No
*13*	J11-7	58	Male	1	Meningothelial	Primary	No	No
*14*	J15-4	40	Female	1	Myxoid	Primary	No	No
*15*	J16-5	49	Female	1	Psammomatous	Primary	No	No
*16*	J17-7	49	Female	1	Not Specified	Primary	No	No
*17*	J18-6	64	Female	1	Meningothelial	Primary	Gamma Knife 36Gy	6 Cycles of TAC Chemotherapy
*18*	J18-7	41	Female	1	Meningothelial	Primary	No	No
*19*	J19-2	44	Female	1	Psammomatous	Primary	No	No
*20*	J19-5	39	Female	1	Meningothelial	Primary	No	No
*21*	J19-6	44	Female	1	Transitional	Primary	No	No
*22*	J19-7	57	Female	1	Meningothelial	Primary	No	No
*23*	J2-2	65	Female	1	Meningothelial	Primary	No	No
*24*	J21-4	55	Female	1	Rhabdoid Morphology Present	Primary	No	No
*25*	J21-5	65	Male	1	Secretory	Primary	No	No
*26*	J21-7	51	Female	1	Fibrous	Recurrent	No	No
*27*	J4-1	63	Female	1	Fibrous	Primary	No	No
*28*	J4-4	67	Male	1	Fibrous	Primary	No	No
*29*	J7-6	57	Female	1	Transitional	Primary	No	No
*30*	J8-3	38	Female	1	Meningothelial	Primary	No	No
*31*	J9-3	70	Female	1	Psammomatous	Primary	No	No
*32*	*JEN	55	Male	1	Psammomatous	Primary	Unknown	No

Tumor sample identifier code with patient age and gender as well as tumor classification variables including histologic subtype, whether the tumor is primary or recurrent, and history of radiation and chemotherapy. University of Utah cell lines are denoted by an asterisk. TAC chemotherapy = Taxotere, Adriamycin, and Cyclophosphamide.

## Data Availability

Data are contained within the article or Appendix A; additional requests for information should be directed to the corresponding authors.

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
