# Peer review of "High-Throughput Mechanistic Screening of Epigenetic Compounds for the Potential Treatment of Meningiomas"

_jcm, 2021, doi:10.3390/jcm10143150_

Round 1

Reviewer 1 Report

That broad range HDACi drugs such as those highlighted in this study often demonstrate potent anti-tumor activity is already know as it has been demonstrated in many other tumor models. In this study, the authors have tried a similar approach to look for FDA approved epigenetic drugs that are cytotoxic to meningioma cells derived from patient tissues. Although the results are promising they are preliminary and require further validation and these points may be addressed before results from this study can be accepted confidently.

  1. The study is essentially based out of only one cell-based assay, MTS assay, as an indicator of cell viability in response to treatment with just one dose of the drugs in the library. Further, the results from this single screen have been analysed and presented in multiple different criteria (based on age, sex, tumor grade, treatment history) stemming from the samples derived from these patients. These results have been presented as six different figures in this study.

Out of the 32 tumors in this study, 25 were grade 1, 6 were grade two and only one was grade 3. Further only 3 were recurrent as opposed to 29 primary. Next, only one was subjected to radiation prior and the ratio of males to females is skewed as well. How confident are the authors thus with their findings, analysis and the claims they have made when their dataset is prone to be affected because of these factors ?

  1. Did the authors think of including other assays instead to confidently support their findings using cultured cells in vitro as understandably the cell amount from patient samples was a limiting factor? MTS assay although widely used as an indicator of cell viability, is essentially an essay to determine the metabolic activity of cells and do not indicate the drug’s ability to merely halt proliferation or cause cell death. Employing FACS cell cycle/apoptosis analysis or trypan clue assay for proliferation with hit compounds would have provided a much better insight into drug mechanism of action. What are the author’s thoughts on this ?
  2. The authors have obtained the results from MTS assay as a result of just a single dose exposure to cells with these drugs. Were there technical replicates in this study? Was the assay performed at least 3 times with each drug independently ? Could the authors provide more clarity here ?
  3. In the MTS assay did the authors just note the absorbance at 490 nm? Did the record the absorbance at 630 nm for background control to subtract and then analyse ?
  4. The authors in their methods have stated they used the drugs at a final concentration of 1 mM and in discussion they state that the concentration was 1 µ Is this a mistake?
  5. One concentration was used for all drugs regardless of the fact that these drugs are vastly differing in their mechanism of action. Did the authors consider the importance of using atleast 2 different concentrations instead? Some drugs which were rendered ineffective or less effective would have merely required a slightly higher concentration? Similarly, those which were effective, could it be because of using a very high concentration 1 mM (provided that is what the authors used) ? What was the vehicle control used and at what concentration, was it uniform across all drugs ? Were there any other controls ? Could the authors clarify these confusions ?
  6. Lastly, if the authors used 1 mM as the final working concentration for drugs in the MTS assay, it is a very high concentration indeed. How does this translate to any therapeutic relevance then since these drugs may only be therapeutically useful to treat meningiomas at a very high concentration in vivo and would potentially lead to more damage in form of side effects than any benefit.

Reviewer 2 Report

The paper is devoted to the interesting and important subject. It is done very well methodologically, and very well presented. Though it does not contain any novelty in methodology, it can be of a high interest to the readers from medical and bio-medical fields due to the data which are presented.

Some comments for its improvement:

1.For me, the description of the components of the 139-compound epigenetic library will be useful for the readers and for the future impact of the paper. For example, the specification of different functional groups of compounds with their representatives will add a big portion of sense to the analysis of the data presented.

2.Does the ORDER of Tumor Samples in Figs. 1-6 are the same as Table 1? If yes, it shoud be written, if not- the names of Tumor Samples should be provided.

Definitely, this order is different in Suppl. Figs, showing different groupings (grades) so all samples must be written on each Fig.

3.The section about Impact of Gender is not clear and the corresponding Fig.6 is not clear. Firstly, the conclusion is not written clearly -there is or there is no Impact of Gender on the Inhibition? The numbers, presented in the text, show that there is not, but the figure seems to be done to show the opposite-? Why the amount of Tumor Samples and the number of effective compounds are different on the Fig?

  1. I would avoid sentences like: “On average, 27.2 +/- 145 15.3SD compounds significantly reduced cell viability”, because compound can not be fractional.

Reviewer 3 Report

The work entitled “High-Throughput Screening of Epigenetic Compounds for the Treatment of Meningiomas”by PD Tatman, TH Wroblewski et al. aim to identify novel therapeutics for the treatment of meningiomas. To this aim, the authors performed a high-throughput drug screen of 32 patient-cultured meningiomas against a 139-compound epigenetics library (Cayman Chemical, Ann Arbor, MI, USA).

This work obtained important results. First, screening of 32 patient-cultured meningiomas demonstrated epigenetic compounds can significantly decrease cell viability.

While all meningiomas displayed a unique sensitivity profile, most were sensitive to a small group of compounds described, including panobinostat, LAQ824, HC toxin, Gemcitabine, JIB-04, and Sb939.

Authors then study the impact of grade, primary vs. recurrent, radiation history, and gender. While interesting and potentially relevant, further experiments are required to confirm some of the findings.

1) Some limitations of the study reside in the single-dose for all the components. The final concentration of the compounds used to perform the screening with the library has to be clarified. In the Methods section (line115), the authors indicate 1 mM for 72h, while in the discussion, the authors indicate 1 µM (line 468) selected based median maximum plasma concentration in humans across a cohort of 145 FDA-approved cancer compounds on reference #39.  More information about how this concentration was selected should be provided.

These doses could be too high for some of the compounds and some effects could be explained, in part, by off-target effects. For example, panobinostat, also named LBH58, has been shown to inhibit the growth of several cell lines at the nanomolar range. (E.g., non-small cell lung cancer cell lines (such as human H1299, L55 and A549 with IC50 of 5 nM, 11 nM and 30 nM, respectively), mesothelioma (such as human OK-6 and Ok-5 with IC50 of 5 nM and 7 nM, respectively) and small cell lung cancer cell lines (such as human RG-1 and LD-T with IC50 of 4 nM and 5 nM, respectively).

As reported here https://www.selleckchem.com/products/LBH-589.html

2) Authors claim that multiple drugs that target HDACs were identified as having the ability to significantly inhibit meningioma growth, suggesting that HDAC inhibition is a promising therapeutic modality for the treatment of meningiomas. Samples were collected from the operating rooms screened within three weeks of surgical resection and never cryogenically frozen, with the exception of tumors originating from the University of Utah (3 lines indicated as  IOMM Lee, GAR and GEN).

They should confirm the sensitivity to panobinostat, LAQ824, HC toxin, Gemcitabine, JIB-04 and Sb939using meningioma cell lines. Authors should estimate the GI50/IC50 of these compounds on IOMM Lee, GAR and GEN and/or other established cell lines to validate the findings.

The unique validation of the findings claimed for the authors make reference to other papers using pan-HDAC inhibitor and Panobinostat in vivo (References #37 and #38)

Comparison between genders, irradiated vs non-irradiated etc should be performed by comparison of GI50/IC50 values. The paper will strengthen by performing other classical culture experiments as colony-forming assays in addition to MTS tetrazolium assays.

3) Authors manifest that the findings implicate that each grade of meningioma has a different sensitivity profile, though all three grades remain sensitive to the small cohort of “broadly effective compounds.” Due to the limited number of samples for grade 3 (n=1), recurrent (n=3) and irradiated (n=1) these data need to be further confirmed before doing some statements. This could be done by using another panel of meningioma cell lines (established).

Minor things

Please add the sample ID in the X-axis of Fig 1B, Fig 2, Fig 3 D,  Fig 4C, Fig 5D, and 6C. If possible, classify/order by tumor grade when comparing primary vs recurrence, radiation treatment or gender (Fig 1B, Fig 2, Fig 4C, Fig 5D and 6C).

Round 2

Reviewer 3 Report

The authors clarified several aspects of the manuscript, including experimental design and dosage. In addition, they explained some of the limitations of the work in the discussion (e.g., number of samples, grade, sex, radiation cohorts, etc.).

The authors present several in vitro and in vivo data that could support, at least in part, the findings of this work. However, the authors indicate that these data are part of a follow-up study and manifest their intention of not including them in this manuscript. Authors claim this paper, if published, will help other labs to establish their protocol so they will answer the questions as a community instead of relying on a single institution´s patient volume.

I still insist that authors should confirm (at least) the sensitivity to panobinostat, LAQ824, HC toxin, Gemcitabine, JIB-04, and Sb939 using meningioma cell lines to confirm the findings of the screening.  Otherwise, the screening lacks an essential validation step. I understand the sample size's limitations, but authors should estimate the GI50/IC50 of these compounds on IOMM Lee, GAR and GEN, and/or other established cell lines to validate the findings. Ideally, authors should use other methods than the MTS assay (cell counting with trypan blue, BrdU, cell cycle analysis, etc.).

Author Response

We would like to thank the reviewer for concerns over the screen's validation; however, it needs to be emphasized that this screen is a mechanistic screen and not a drug candidate screen.  We would argue that the validation process is quite different between the two approaches. The purpose of this study is to find a common mechanism that could be targeted for the treatment of meningiomas; it is not, and should not, be interpreted as a study that promotes a specific drug candidate. For instance, as a class of HDAC inhibitors, panobinostat has poor blood brain barrier penetration and would likely not be worth the effort, time, or expense to validate any further. The fact that multiple HDAC inhibitors were “hits” does support HDAC inhibition as a potentially viable treatment motif for meningiomas, and to investigate this a second screen and follow-up studies to identify targets are needed. Subsequently, we have performed a second screen looking specifically at HDAC inhibition that covers most families of HDACs; there, we identified romidepsin as the most promising compound for further validation. At this stage, candidate validation is necessary, and your line of suggested experiments would then come into play, with the gold standard being demonstrating efficacy in vivo., Indeed, we have demonstrated that romidepsin inhibits meningioma growth in xenograft mouse models. All this information is part of the follow up study to the current publication, and we have made these unpublished data available to you in our prior response. That information included the data to calculate IC50s for all the HDAC inhibitors in the screen across dozens of tumors, as well as the in vivo mouse data.  The efficacy of romidepsin in vivo provides better validation as a compound (and as a class of drugs) than evidence any in vitro experiment could generate. At this time, we believe there is no reason to perform validation for panobinostat, or any of the other research compounds in the screen, given the strength of the work we are finishing with romidepsin, which we have confidentially shared with you. In this current manuscript, we show that screens for epigenetic compounds could select for classes of targets, in this case, HDACs. In our follow-up manuscripts, we will

we will show that HDAC inhibition is effective in vitro and in vivo, which will include IC50 derivations, cell death mechanisms, and relevant in vivo tumor growth studies. We are sorry for the continued misunderstanding and miscommunication about our data, and have added portions to the Introduction and the Discussion to highlight the nature of this mechanistic screen.